# Crop Cycle and Tillage Role in the Outbreak of Late Wilt Disease of Maize Caused by *Magnaporthiopsis maydis*

**DOI:** 10.3390/jof7090706

**Published:** 2021-08-28

**Authors:** Ofir Degani, Asaf Gordani, Paz Becher, Shlomit Dor

**Affiliations:** 1Plant Sciences Department, Migal Galilee Research Institute, Kiryat Shmona 11016, Israel; asigordani1@gmail.com (A.G.); pazbec@gmail.com (P.B.); dorshlomit@gmail.com (S.D.); 2Faculty of Sciences, Tel-Hai College, Tel-Hai 12210, Israel

**Keywords:** *Cephalosporium maydis*, crop rotation, crop protection, disease management, fungus, *Harpophora maydis*, late wilt, maize, real-time PCR, rhizosphere

## Abstract

The destructive maize late wilt disease (LWD) has heavy economic implications in highly infected areas such as Israel, Egypt, and Spain. The disease outbreaks occur near the harvest, leading to total yield loss in severe cases. Crop rotation has long been known as an effective means to reduce plant diseases. Indeed, agricultural soil conservation practices that can promote beneficial soil and root fungi have become increasingly important. Such methods may have a bioprotective effect against *Magnaporthiopsis maydis*, the LWD causal agent. In this two-year study, we tested the role of crop rotation of maize with either wheat or clover and the influence of minimum tillage in restricting LWD. In the first experiment, wheat and clover were grown in pots with LWD infected soil in a greenhouse over a full winter growth period. These cultivations were harvested in the spring, and each pot’s group was split into two subgroups that underwent different land processing practices. The pots were sown with LWD-sensitive maize cultivar and tested over a whole growth period against control soils without crop rotation or soil with commercial mycorrhizal preparation. The maize crop rotation with wheat without tillage achieved prominent higher growth indices than the control and the clover crop cycle. Statistically significant improvement was measured in the non-tillage wheat soil pots in sprout height 22 days after sowing, in the healthy plants at the season’s end (day 77), and in shoot and cob wet weight (compared to the control). This growth promotion was accompanied by a 5.8-fold decrease in pathogen DNA in the plant stems. The tillage in the wheat-maize growth sequence resulted in similar results with improved shoot wet-weight throughout the season. In contrast, when maize was grown after clover, the tillage reduced this parameter. The addition of commercial mycorrhizal preparation to the soil resulted in higher growth measures than the control but was less efficient than the wheat crop cycle. These results were supported by a subsequent similar experiment that relied on soil taken from commercial wheat or clover fields. Here too, the wheat-maize growth cycle (without permanent effect for the tillage) achieved the best results and improved the plants’ growth parameters and immunity against LWD and lowered pathogen levels. In conclusion, the results of this study suggest that wheat and perhaps other crops yet to be inspected, together with the adjusted tillage system, may provide plants with better defense against the LWD pathogen.

## 1. Introduction

The importance of maize (*Zea mays* L., corn) cannot be over-emphasized, especially in developing countries [1]. Maize is produced annually more than any other grain, reflecting its importance globally [2]. Late wilt, a disease severely affecting maize fields throughout Israel [3], is characterized by rather rapid dehydration of the plants near maturity. It is considered the most harmful disease in commercial maize fields in Israel [4] and Egypt [5] and poses a significant threat in India [6,7], Spain, and Portugal [8]. The disease is gradually continuing to spread and is currently reported in at least eight countries. The causal agent is the fungus *Magnaporthiopsis maydis* [9], recognized by two additional synonyms, *Cephalosporium maydis* [10] and *Harpophora maydis* [11,12].

The pathogen can survive in the soil for long periods. When a susceptible host plant is seeded, the fungi can penetrate the plants’ roots, causing root necrosis and affecting sprout development [13,14]. First aboveground symptoms usually appear later in the season as plants begin to flower and are enhanced under drought conditions [15,16]. When the growth session advances, *M. maydis* spreads upwards inside the plants’ vascular system, disrupts the water supply and leads to dehydration [17]. In heavily infested fields planted with sensitive maize hybrids, late wilt may cause 100% infection and total yield loss [18]. If ears are produced, the kernels that do form are poorly developed and are infested with the pathogen.

*M. maydis* can survive and spread through seeds [19], infested soil, crop residues [20], or secondary hosts such as lupine [21], cotton [22,23], watermelon, and *Setaria viridis* (green foxtail) [24]. Various LWD prevention methods have been examined over the years, and some gained positive results in reducing LWD in commercial fields. These include balanced soil fertility [25,26], adjusted tillage system and cover crop [27], watering the field [28], biological approaches (will be discussed in detail below), soil solarization [29], allelochemical [13], and chemical options [4,30,31,32,33]. However, none of these methods is currently being used in Israel. Instead, worldwide LWD is controlled by more economically effective management by developing genetically resistant maize cultivars [5,6,34].

The National Maize Program in the Agricultural Research Center in Giza, Egypt, identified many sources of resistance; the release of resistant cultivars since 1980 has significantly reduced late wilt losses in Egypt [35]. A breeding program for resistant germ lines has been operational in Israel for about a decade (Israel Northern R&D, Migal–Galilee Research Institute, Kiryat Shmona, Israel). However, the presence of highly aggressive isolates of *M. maydis* [36,37] may threaten these resistant maize cultivars. In addition, the pathogen could spread in relatively resistant plants that showed no symptoms. Infected seeds, even those of non-symptomatic plants, can spread the disease [3,19].

In 2017–2018, the search for a chemical application to control LWD led to an economically feasible solution [4,18,31] in Israel. The successful treatment protocol is based on changing the maize cultivation method, changing the traditional irrigation method used in most corn-growing areas in Israel, and the sophisticated integration of Azoxystrobin-based pesticide mixtures in a schedule adapted to key points in the development of the disease.

Still, it was shown earlier that the pathogen is present in the host tissues of successfully chemically treated plants [32]. This finding hints at the potential risk that the pathogen will develop immunity to Azoxystrobin, the most effective antifungal compound against the late wilt pathogen [4,18]. Unfortunately, the rapid development of resistance to this fungicide and the consequential control failure in many crops has become increasingly problematic [38].

The growing trend of reducing pesticide use [39] raises the need for alternative ways of coping with severe fungal diseases such as the late wilt of maize. Indeed, biological eco-friendly control methods to restrict *M. maydis* and other phytopathogens are at the forefront of the current scientific effort. Two environmentally friendly strategies to control late wilt are presently in this scientific focus.

First, the use of *Trichoderma* sp. and other protective microorganisms as a biocontrol agent has been demonstrated in the past in culture media, in greenhouse plants, and in the field with very promising potential (most recently [40,41]). Likewise, we previously conducted two years of research with new *Trichoderma* species: *T. asperelloides* (T.203); *T. longibrachiatum* (T.7407 from marine source [42]); and *T. asperellum* (P1), an endophyte isolated in our laboratory from corn seeds of a strain susceptible to LWD [43]. These isolates prevented the pathogen’s growth in culture plates, significantly reduced its establishment and development in seedlings’ corn plant tissues, and resulted in significant improvement in growth and crop indices under field conditions [16,44].

Second, maintaining soil mycorrhizal fungi between seasons has proven to be an essential factor towards the same goal (summarized by [40]). Results from other phytopathogens suggest that under no-till cropping, selected cover crops or crops in a rotation could help build mycorrhizal communities that function throughout a sequence of several main crops [45]. Aggressive tillage combined with long periods where the field is unprocessed results in the destruction of the integrity of mycorrhizal networks. Maintaining the continuity and integrity of mycorrhizal networks in the soil may allow the plant to enjoy higher resistance to soil diseases [46], including late wilt disease [27]. Indeed, Arbuscular mycorrhizal fungi (AMF) have a proven ability to improve plant resistance to biotic and abiotic stresses by activated the plant’s local and systemic defense mechanisms [47].

So far, this approach of strengthening the soil microbiome was poorly tested against the late wilt pathogen, *M. maydis*. This knowledge gap is now encouraging the exploration of this method’s potential as part of an integrated control program to restrain the late wilt agent, as we will detail below. In the current study, this method was studied thoroughly and evaluated over two years. In the first year, the effect of crop rotations of wheat/maize or clover/maize and a simulation of tillage regimes (no-tillage or conventional tillage) was inspected in pots where LWD-susceptible maize cultivar was grown in infected soil. These treatments were compared to infected no-tillage soil and infected no-tillage soil + commercial mycorrhizal preparation (Resid MG [48], BioBee, Sde Eliyahu, Israel). In the second year, the field’s soils after commercial cultivation of wheat, clover, or soil without previous winter growth were used. These soils underwent two different tillage regimes before maize cultivation—no-tillage or conventional tillage. The effectiveness of these practices on *M. maydis* pathogenesis was studied by evaluating the growth parameters throughout the season, estimating the disease symptoms, measuring yield production, and monitoring the pathogen’s DNA inside the host tissues using quantitative real-time (qPCR)-based approach.

## 2. Materials and Methods

### 2.1. Fungal Species and Growth Conditions

The *M. maydis* isolate *Hm-2* (CBS 133165, CBS-KNAW Fungal Biodiversity Center, Utrecht, The Netherlands) was isolated from diseased maize plants collected from Sde Nehemia (Hula Valley, Upper Galilee, northern Israel) in 2001 and identified as previously described [3,49]. The fungus was grown on solid, rich potato dextrose agar (PDA) (Difco, Detroit, MI, USA) medium in the dark at 28 ± 1 °C for 4–7 days before being used.

### 2.2. The Preparation of Infected Sterilized Wheat Grains

Infected sterilized wheat grains were used to spread the fungus in the soil, as previously described [50]. Wheat grains were soaked overnight in tap water. The grains were then dried for about four hours in a fume hood on paper towels and autoclave sterilized for 30 min at a temperature of 120 °C. Disinfected plastic 0.5 L boxes were used to inoculate 100 g sterilized wheat grains with 10 *M. maydis* mycelial discs. Mycelia disks (6 mm in diameter) were taken from the margins of a 4–6-day-old fungal colony grown as described in Section 2.1. The boxes were sealed with a lid (that was tightened to the box using Saran wrap), covered with aluminum foil (to guarantee dark conditions), and incubated at 28 ± 1 °C in the dark for 10 days.

### 2.3. The Plant Inoculation Methodology

The plant inoculation technique and maize growth were similar to that of Degani et al., 2019 and 2020 [18,22]. The inoculum method was based on the use of soil from a field, having a long history of LWD infection (Amir, Mehogi-1 plot, coordinates: 33°09′59′′ N 35°36′52′′ E) [4,18] and a complementary inoculation with the Hm-2 isolate that was carried out in two steps. First, 40 g of sterilized infected wheat seeds were mixed with the top 20 cm of each pot’s soil one week before seeding the winter crops (wheat and clover). The same procedure was done in the control pots as well. These seeds were prepared as described in Section 2.2. Second, with the maize plants’ aboveground emergence (in the summer season, 8 days after sowing, DAS), two *M. maydis* colony agar disks (6-mm-diameter, see Section 2.1) were added to the upper parts of the roots (4 cm beneath the ground surface).

### 2.4. Crop Rotation and Tillage Effect in Pots over a Full Growing Season—The 2019 Experiment

This study examined the effect of crop alternation and tillage on *M. maydis*’ ability to cause LWD over a whole growth period in pots in an open greenhouse. A binary rotation (with phased entry) of wheat–maize or clover–maize was examined during the fall and the winter of 2019 until the summer of 2020 at Tel-Hai College, Upper Galilee, Northern Israel. Instead of experimenting in the field, the use of pots with naturally infested soil aimed at enhancing the soil inocula and achieving high and equable infection as much as possible. The use of pots also ensures providing a better isolate of each treatment’s influence and controlling the water regime. It should be emphasized that pathogenicity experiments cannot rely upon natural soil infection alone, resulting in highly variable data. Even in a heavily infested area, the spreading of the pathogen is nonuniform [18,51]. The pathogen is dispersed in small quantities in the soil, and the disease spread is not uniform in the field.

The environment data recorded during the growing season were nearly optimal for the winter crops, maize growth season, and LWD burst, as we will specify below. Sixty 10 L pots were filled with local naturally infested peat soil as described above (Section 2.3). The soil was mixed with Perlite No. 4 (for the ground aeration) at a ratio of 2:1. The pots were placed on a metal table made of mesh to prevent the passage of roots to the ground. Each treatment included 10 independent replications (pots) and had two controls and four treatments (Figure 1):

(1) Infected control—naturally infested soil with the addition of complementary *M. maydis* infection (as described above). The pots with soil were kept all winter without a crop, and no-tillage was applied before the maize seeding in the spring.

(2) Commercial control—similar infected soil (without winter cropping or tillage) with the addition of commercial Resid MG [48] (or Bio-Up) product (Symborg S.L., Murcia, Spain, supplied by BioBee Biological Systems, Sde Eliyahu, Israel). This product contains 1.6 × 10^4^ spores/kg of the Arbuscular mycorrhizal, *Glomus iranicum* var. *tenuihypharum*, on clay particles. The preparation was added according to the manufacturer’s instructions: 5 g of the product powder was added to the sowing pothole before sowing the corn.

(3,4) Wheat–maize crop rotation–similar infected soil after the wheat growth season without tillage or with a simulation of conventional tillage. The last treatment was conducted using vigorous mixing and disintegration of the soil by pickaxe.

(5,6) Clover–maize crop rotation–similar infected soil after the clove growth season with or without tillage.

#### 2.4.1. Winter Season

The wheat cultivar Zahir was selected as a winter crop for this study. This hybrid is common in commercial fields in Israel due to its being suited for semi-desert conditions and rich in crops even under conditions of low rainfall. The clover cultivar was Tabor, an annual, single-harvest cultivar that is very popular among growers in Israel since it is suitable for most soils. Both wheat and clover cultivars were supplied by Hazera Seeds Ltd., Berurim MP Shikmim, Israel. The winter crops were seeded on December 11, 2018, and the aboveground parts were harvested (with minimum intervention in the soil integrity) on 29 April 2019, 139 DAS. The clover pots were treated after harvest with Carfentazone ethyl (kill broadleaf plants with no effect on Poaceae, Or, Tapazol Chemical Works Ltd., Beit Shemesh, Israel). The meteorological data recorded during the winter growing season were: temperature 12.9 ± 5.1 °C, humidity 78.6 ± 20.7%, radiation 147.5 W/m^2^, precipitation 1566 mm, and evaporation 424.0 mm (data—average ± standard deviation according to the Israel Northern Research and Development meteorological station data, Hava-1). The irrigation commenced eight days later, five days before the beginning of the maize season.

#### 2.4.2. Summer Season

On 12 May 2019, 13 days after the harvest of the winter crops, each of the pots was sown with five seeds of the maize Prelude cv. (a sweet variety, produced by SRS—Snowy River Seeds Australia, marketed by “Green 2000,” Israel). This strain had been tested in the past and was shown to be highly susceptible to LWD [18,50]. Seeds were pre-treated with thiram, captan, carboxin, and metalaxyl-M (manufactured by Rogers/Syngenta Seeds, Boise, ID, USA, supplied by CTS, Tel Aviv, Israel), a standard general pesticide treatment. The seeds were tested for vitality before sowing and were buried in the ground at a depth of 4 cm. The pots were marked and randomly scattered. The maize plants were harvested on 28 July 2019, 77 DAS. The meteorological data recorded during the growing season were nearly optimal for the LWD burst: temperature 26.8 ± 6.2 °C, humidity 55.3 ± 12.6%, radiation 359.7 W/m^2^, precipitation 0.5 mm, and evaporation 658.4 mm (data—average ± standard deviation according to the Israel Northern Research and Development meteorological station data, Hava-1). Since it did not rain during the corn growing season, watering was carried out from seeding using 2 L per day per pot. Fertilizers and insecticide treatments were applied according to the Israel Ministry of Agriculture Consultation Service (SAHAM) growth protocol.

After eight days, the germination percentages were calculated. At 22 DAS, the sprouts were thinned to one plant per pot, and the thinned plants were used to evaluate the plants’ growth indices. On the female flowering day (56 DAS), the phenological stage and height of the plants were assessed. At the experiment’s end (77 DAS), the plants’ health status, height, fresh roots and shoot weight, and yield (total cob weight) of the treatments were measured. Wilt assessment was based on four categories related to the whole plant: healthy (1), minor symptoms (2), dehydrated (3), and dead (4). In addition, samples from the first internode of the plant stems were taken for DNA purification and qPCR.

### 2.5. Evaluation of Commercial Field Soil after Wheat or Clover Growth—The 2020 Experiment

The 2020 experiment was conducted according to the same experimental design as in the 2019 experiment throughout a full growing season. This includes the same wheat, clover, and maize cultivars and the same soil inoculation protocol. The 2020 trial included four treatments and two controls and was performed in 10–12 repetitions—a total of 113 pots. It was conducted in a closed greenhouse on an experimental farm (R&D North Israel, Kiryat Shemona, Israel) located in the Upper Galilee, Hula Valley, northern Israel. Commercial field soil after the winter growing season of clover and wheat, and soil after the winter without early growing (the last crop grown was watermelon in the last summer season) were taken from the fields of Moshav Moledet (Gilboa Regional Council, Lower Galilee, Israel). The soils were divided, and simulation of conventional tillage or no-tillage was applied to the 10 L pots. The simulation of conventional tillage treatment was conducted by vigorous mixing and disintegration of the soil by a hoe before transferring it to the pots. The no-tillage treatment was performed by carefully digging and moving the field soil to the pots as one block in an attempt to maintain soil integrity as much as possible.

The pots were placed on stone blocks to prevent the passage of roots to the ground (Figure 2). Each of the pots was filled with water until saturation one week before maize sowing. After this soil initiation, the soils were inoculated with *M. maydis*—the late wilt disease causal agent, as described in Section 2.3. The pots were sown with the susceptible maize hybrid Prelude cv. on 10 August 2020. The average temperature was 33 ± 12.0 °C, and the average humidity was 27 ± 7.4%. The pots were watered with approximately 2 L per day per pot and received fertilizers and treatments against various pests according to the Israel Ministry of Agriculture Consultation Service (SAHAM) instructions. The plants were harvested on 20 October 2020, 71 DAS.

Emergence percentages evaluation was made 3 DAS. After 43 days, the sprouts were diluted into one plant per pot. On the day of dilution and at the end of the experiment on day 71 of growth, the following indices were examined: plant height, shoot weight, phenological stage, and signs of dehydration. Seasonal variations of the main experiment stages are demonstrated in Figure 2. These stages include the sowing, the end of the sprouting phase (30 DAS, V5-fifth leaf stage, before the female flowering at 56 DAS), and near the harvest (71 DAS). The maize development was normal and correlated to the data in the literature (see [52]).

The dehydration symptoms on the large bracts surrounding the cobs (the spathes) were estimated, as previously described by [16], according to the following scale: 1—healthy with no signs of dehydration, 2—minor symptoms up to 20% of the cob surface, 3—moderate symptoms that cover 30–40% of the cob surface, and 4—diseased with 50% or more dehydrated cobs’ area. In addition, DNA was extracted from the tissues of 10 plants for treatment, and a qPCR test was made to detect *M. maydis* DNA in the plant tissues (roots on the day of thinning and root and shoot on the harvest). At the end of the experiment, crop evaluation was also performed.

### 2.6. Molecular Diagnosis of the Late Wilt Pathogen

*M. maydis* target DNA qPCR detection was made for maize plants from all 10 pots of each treatment. Samples were taken from the roots and lower stem (near the first internode) tissues. Plant tissues were washed vigorously under running tap water to remove any visible soil. Tissue sampling was made by cutting a 20 mm (in length) piece from each plant’s lower stem near the aboveground first internode. One repeat was considered as 0.7 g tissue fresh biomass. A 4 mL CTAB buffer was added to each plant sample, and the blend was moved into universal extraction bags (Bioreba, Switzerland). The tissue was processed for 5 min using a hand tissue homogenizer (Bioreba, Switzerland). Then, DNA purification and qPCR-based quantification were conducted according to [31]. The A200a primers were used for qPCR (*M. maydis* amplified fragment length polymorphism-derived species-specific fragment) [53,54]. The gene cytochrome c oxidase (*COX*) encoding the eukaryotic mitochondria respiratory electron transport chain, the last enzyme, was a reference “housekeeping” gene used to normalize the amount of DNA [55,56]. The relative gene abundance calculation was made according to the ΔCt model [57]. Similar efficacy was assumed for all samples. All amplifications were performed in triplicate.

### 2.7. Statistical Analyses

A fully randomized statistical design was used in the field observation preparation. Data analysis followed by statistical analysis was made using the JMP program, 15th edition, SAS Institute Inc., Cary, NC, USA. All the data in this manuscript were subjected to the same analytical method. The one-way analysis of variance (ANOVA) tracked by multiple comparisons post hoc of the Student’s *t*-test for each pair (without multiple comparisons correction) was used to estimate the differences between the treatments and between the treatments and the controls. In these growth experiments, achieving a uniform infection with *M. maydis* in Israeli strains is challenging due to natural variations in fungi pathogenesis [18,50]. Thus, the molecular DNA measurements resulted in a high level of variations within the results (high standard error values), and statistically significant differences could hardly be identified.

## 3. Results

Strengthening soil microbiome integrity by avoiding aggressive tillage and encouraging specific communities of microorganisms that inhabit a plant’s surroundings and roots may enhance soil immunity against plant-parasitic fungi such as the LWD agent, *M. maydis*. To deepen our understanding towards this end, we conducted two experiments over the 2019 and 2020 winter to summer growth periods. The study was performed as a dual crop with a time interval of 1–3 weeks between crops with and without invasive tillage of the soil. We selected wheat and clover as the alternative winter crops to be rotated separately with maize. These two winter crops are prevalent in Israel and are common in the crop cycle with maize in Israel and worldwide.

### 3.1. Crop Rotation and Tillage Effect in Pots over a Full Growing Season—The 2019 Experiment

The dual cultivation greenhouse experiment was conducted during an entire growth period in the winter (wheat and clover season), spring and summer (maize season) of 2019. All treatments had a positive effect on the emergence of the maize plants, evaluated 8 DAS, compared to the control (Table 1). Significant statistical difference (*p* < 0.05) in this measure was found in the commercial mycorrhizal treatment (Resid MG) and the wheat and clover soils after tillage.

At 22 DAS, a mixed trend was observed (Table 1). The commercial mycorrhizal treatment significantly improved sprout height but did not affect the other measures. The clover soil without tillage positively and significantly affected shoot weight (without improvement in the other criteria), but this soil’s tillage abolished this achievement. In wheat soil, the tillage improved shoot weight but abolished the significant improvement in sprout height recorded in the undisrupted ground. Interestingly, 34 days later, at the fertilization (56 DAS, Table 2), the tillage accelerated the clover soil plants’ development (into the R1 stage) and improved plant height in both the clover and wheat soils.

At harvest (77 DAS), maize plants that were grown after wheat acquired significant resistance to late wilt disease, which had not been present in the other treatments (Figure 3). This resistance was affected mainly by an elevation in the healthy plants’ proportion, from 10% in the control to 80% in the undisturbed wheat soil. The wheat soil immunity to late wilt is expressed in growth promotion (Table 3), whereas the highest values were received in shoot weight (significant in both the non-tillage and the tillage soil) and cob weight (significant in the non-tillage soil). Since the dry weight values were similar to all treatments and the control (without statistical significance, Table 4), variations in the plants’ fresh weight among the treatments resulted from differences in the water content level rather than in the tissues biomass. The non-tillage wheat soil also led to 5.4-fold reduced DNA of *M. maydis* within the plants’ lower stalk (Figure 4). When commercial mycorrhizal preparation was applied to the ground prior to the maize growing season, an increase in growth parameters was evident compared to the control (Table 3). Yet, this increase was only significantly different.

### 3.2. Evaluation of Commercial Field Soil after Wheat or Clover Growth—The 2020 Experiment

This follow-up experiment was conducted using the same design as the 2019 experiment. Still, instead of performing a clover and wheat winter growing season in the pots, a commercial field soil of those crops was used. The sprouts’ emergence percentages at 3 DAS growth showed a decreasing tendency in the control and clover soil due to the tillage (Table 5). On the other hand, the wheat soil was not affected by tillage in this criterion. The shoot weight values evaluated 43 DAS showed the same tendency, whereas the other growth parameters (the number of leaves and plant height) were only slightly affected by the treatments at that age. The disease severity evaluated by the pathogen DNA spreading in the roots 43 DAS supported the shoot weight values, with the lowest DNA levels in the wheat soil treatment (Figure 5).

At the experiment’s end (71 DAS), the fresh shoot weight improved after tillage in the control and wheat soils but was reduced in the clover tillage soil (Table 6). All other growth indices were affected less by pre-cropping or land processing, except for cob weight, which significantly decreased in the wheat soil with tillage.

The disease symptoms (Figure 6) and pathogen DNA in the roots and shoots of the plants (Figure 7) evaluated on the harvest day (71 DAS) indicate a positive effect of the wheat soil (with improvement after tillage). This LWD depression was expressed by a significant 2.3-fold decrease in the cobs’ dehydration symptoms and a 25-fold and 225-fold reduction in the pathogen’s DNA in the plants’ roots and shoots, respectively.

## 4. Discussion

The rhizosphere, a soil layer adjacent to the root’s surface, is affected by both the presence of a plant and soil properties. This layer has a critical impact on a plant’s existence. A significant part of the roots and the whole plant function depends on the “nature of the rhizosphere” and the biological activities within it. Intensive agriculture causes a decrease in microbial biomass in the soil that causes, over time, a decline in soil fertility and yield [58]. Arbuscular mycorrhizal fungi (AMF), which interact with plant roots and other soil fungi, have known benefits in nourishing plants and conferring disease resistance. These properties make them a valuable tool in modern agriculture [48]. The fungus provides the plant with nutrients, affects root morphology, and improves water balance, while the plant provides it with photosynthesis products. These “natural fertilizers” are obligatory symbionts belonging to the phylum *Glomeromycota* and inhabit 80% of terrestrial plants [59]. Mycorrhiza has an aggressive antagonism with various plant pathogens and may also be used as a biological pesticide [60].

To date, the use of mycorrhizal fungi to protect field crops is limited due to the high costs (compared to chemical fertilizers) and low biodiversity of the commercial applications offered [59]. This is in addition to the long time required from applying the fungus to achieving efficiency, and in light of the fact that such applications often do not match the intensive growing systems used in agricultural fields. At the same time, mycorrhizal fungi have a wide range of plant hosts. Moreover, using the intact extraradical mycelium method (ERM, preserving the continuity of the mycorrhizal network in the soil) makes it possible to produce a crop sequence that will maintain the mycorrhizal network that promotes the plants’ growth and assist in protecting them against soil diseases [61].

Although the worldwide scientific effort is focusing on seeking solutions to LWD based on eco-friendly biological approaches [16,40,41,43,44,62], there is a lack of information on maize performance under LWD stress in crop rotation and reduced tillage. One study that examined this recently [27] showed that grain production and *M. maydis* presence were significantly reduced when both cover crop and minimum tillage were applied together. It was also found that with cover crop and minimum tillage, the arbuscular root colonization was higher.

The current study did not directly examine the effect of preserving and establishing the mycorrhizal network in the soil to deal with late wilt disease in corn. Instead, it focused, for the first time in Israel, on an agricultural practice based on preserving soil microflora integrity (by avoiding tillage) and affecting its nature (by cultivating selected crops in a dual-season growth). A comparative analysis of the 2019 and the 2020 experiments results at the harvest day is illustrated in Table 7.

When corn was sown on wheat soil, a significant improvement in the fresh weight of the shoot (147–154% in the 2019 experiment) and cob (136–146% in the 2019 experiment) was achieved compared to the control. This result was also better than the other treatments (clover soil and commercial mycorrhiza preparation). This achievement was not affected drastically by tillage. It was followed by a sharp decrease in disease symptoms (73% in the 2019 experiment) and the pathogen’s presence (82–64% in the 2019 experiment) in the plants’ tissues (Table 8).

Interestingly, the 2020 experiment that was based on commercial fields soil resulted in more severe LWD. The disease severity is reflected in higher diseased plants percentages and sharp (23-fold) elevation in the pathogen DNA in the first aboveground internode of the no-tillage control plants (Figure 7, Table 8). This is most likely the reason for the lesser enhancement in the shoot and cobs fresh weight in the clover and wheat treatments (compared to the control) this year (Table 7).

Another intriguing aspect is that at the season-ending, the roots appeared comparatively less affected by the pathogen than the shoot (Figure 7). Indeed, *M. maydis* high DNA levels in the shoot at this growth stage (the harvest) are well documented in our previous works [3,4]. When an LWD susceptible maize cultivar was seeded on *M. maydis* heavily infested commercial field soil, the following changes in the pathogen’s DNA levels in the host tissues were recorded. When early signs of the disease began to appear from day 50 onwards, the fungal DNA decreased in the roots but increased significantly in the stems until it peaked at 3.2-fold its initial level 72 d after sowing [3]. These variations in the DNA were consistent with previous literature observations on the disease’s mode and the pathogen spread from the roots to the stem, leaves, and kernels [63].

Thus, it appears that since wheat and maize are more closely related than clover and maize (they are both Poaceae), they may share similar mycorrhizal networks that are adapted to perform better with these crops. Indeed, in the clover–maize sequence, the above-mentioned growth promotion and LWD resistance were reduced, even more so when tillage was applied.

To support this idea, it was reported that crop plants acquired a mycorrhizal fungal community closely related to that of the previous host plant and different from that found when the soil was disturbed or not cropped before growth [45]. In this last study, wheat grown in undisturbed soil immediately after the legume *Ornithopus compressus* acquired a mycorrhizal fungal community closely related to that of the last plant host and different from that found when the soil was disturbed or not cropped prior to the growth of the wheat. Parallel effects were seen in the succession from Poaceae (*Lolium rigidum*) to Fabaceae (*Trifolium subterraneum*), indicating that these effects are not unique to the legume-wheat sequence.

With the novelty of the current study, it is essential to point out that these trials were conducted over one season, while crop rotation may have a long-term effect on soil fungus populations, which may only be evident after a more extended period. A long-term study on the impact of different crop rotations (including wheat–maize) and tillage regimes (no-tillage or conventional tillage) on microbial biomass and other soil properties was established in 1976 in southern Brazil [58]. The no-tillage system showed increases in microbial biomass percentage at the 0- to 5-cm depth. Reduction of tillage had a more significant effect on microbial biomass than crop rotations, particularly in the 0–5 cm depth. These results provide evidence that tillage or crop rotation affects microbial immobilization of soil nutrients. The results of this and other studies indicate that no single cropping system is favored for all fungi [64]. Thus, a tailored solution that will address the crops in the rotation, the tillage system, the cultivar that should be protected, and the pathogens/disease stresses should be carefully planned. Such situations should be examined in subsequent studies. Closely related cultivars such as wheat and maize may benefit from the same control strategy.

## 5. Conclusions

The selection of certain plant sequences under standard environmental conditions may lead to the suppression of weeds, insects, and diseases, and the avoidance of yield decline. The current study was performed as a dual crop experiment involving a time interval of one to three weeks between crops with and without invasive tillage of the soil. The LWD was affected differently by the two rotation systems applied here. The effect of crop rotation on the *M. maydis* spreading in the host tissue, plant growth parameters, and LWD symptoms in the wheat–maize sequence was significant. In the wheat soil, all these parameters improved (*p* < 0.05). These results were obtained regardless of the tillage system applied. In contrast, this positive outcome was less prominent in the clover–maize sequence and was reduced even more so when tillage was used. In environments where *M. maydis* is known at the outset, the choice of cultural conservation practices is essential for maize cultivation, namely, crop cycle, installation of a cover crop, and minimum tillage system. Such green methods could significantly reduce disease pressure and hence assist in reducing the use of fungicides, which have adverse effects on human health and the environment.

## Figures and Tables

**Figure 1 jof-07-00706-f001:**
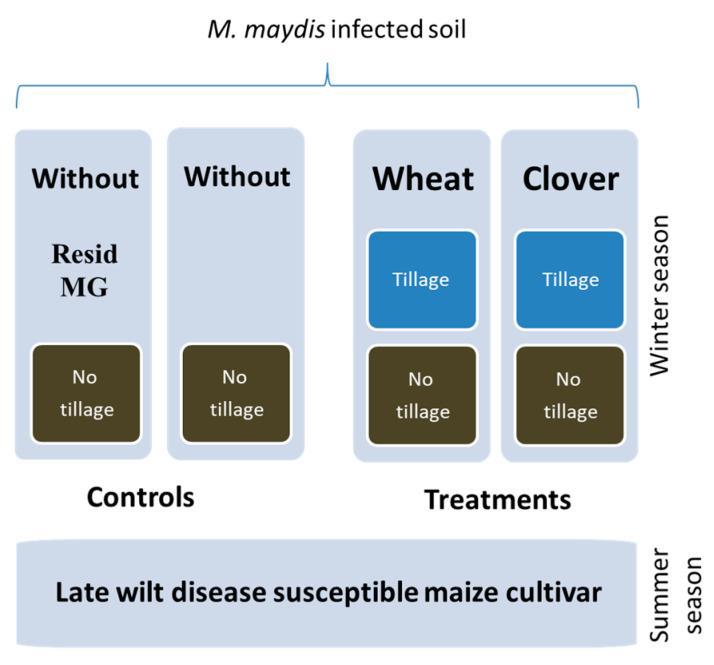
The 2019 experiment program. The experiment was conducted in pots in an open greenhouse over an entire growing season (winter season followed by spring–summer season). The late wilt susceptible maize genotype Prelude cv. was grown on *M. maydis*-infected soil in which wheat or clover had been grown as a crop prior to maize in this dual cultivation. The controls included the same prior crop soil but with tillage or soil that had not been cropped before the maize growth (control). Additionally, for comparison, the commercial Resid MG product of Arbuscular mycorrhizal, *Glomus iranicum* var. *tenuihypharum* (Symborg S.L., Murcia, Spain, supplied by BioBee Biological Systems, Sde Eliyahu, Israel) was tested as an addition to the soil that had not been cropped before growth.

**Figure 2 jof-07-00706-f002:**
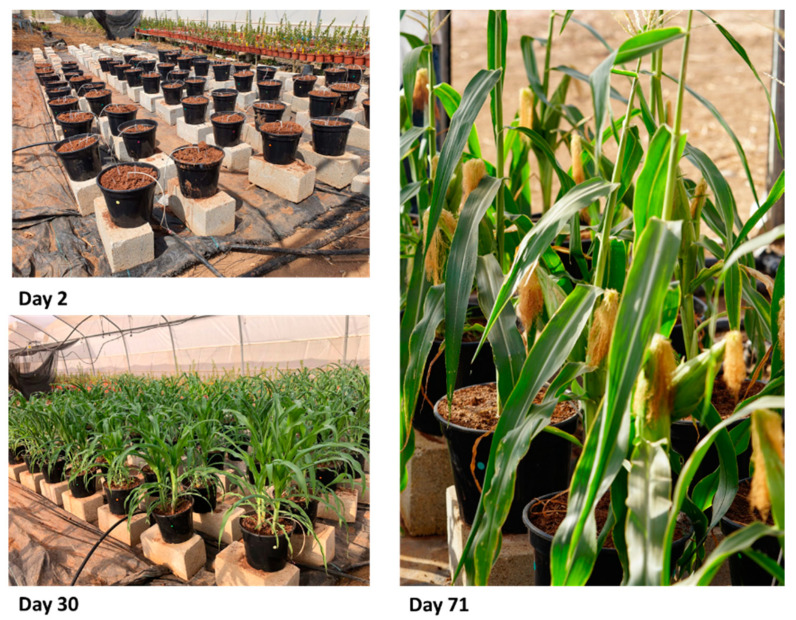
The 2020 experiment photo. The study was performed in 10–12 replications of six treatments in pots in a greenhouse throughout a full maize growing season and examined the effect of pre-growth of winter crops (clover or wheat) and maintaining soil integrity (by avoiding tillage) on late wilt disease. Commercial field soil after the winter growing season of clover and wheat and soil after the winter without early winter growing were used. The soils were divided, and simulation of conventional tillage or no-tillage was applied before moving the field soil to 10 L pots. The soils were inoculated with *M. maydis*, the late wilt disease causal agent, and sown with the susceptible maize hybrid Prelude cv.

**Figure 3 jof-07-00706-f003:**
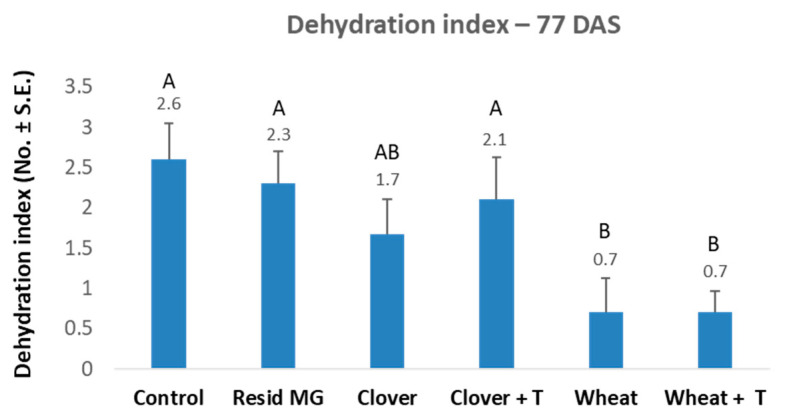
The 2019 soil microbiome integrity experiment–dehydration assessment at the experiment’s end (77 DAS). The experiment is described in Figure 1. The controls included the same prior crop soil but with tillage (+T) or soil that had not been cropped before the maize growth (control). Additionally, similar control with commercial Resid MG product (Arbuscular mycorrhizal, *Glomus iranicum* var. *tenuihypharum*) was tested. Wilt assessment was based on four categories related to the whole plant: healthy (1), minor symptoms (2), dehydrated (3), and dead (4). Vertical upper bars represent the standard error of the mean of 10 replications. Statistically significant (one-way ANOVA, *p* < 0.05) differences between treatments at the same measures are indicated by different letters (A–B).

**Figure 4 jof-07-00706-f004:**
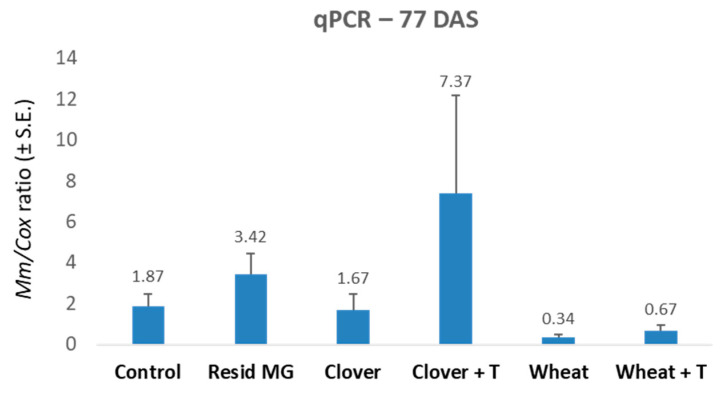
The 2019 soil microbiome integrity experiment—real-time PCR analysis at the experiment’s end (77 DAS). The experiment is described in Figure 1. The *y*-axis indicates *M. maydis* relative DNA (*Mm*) abundance in the roots normalized to cytochrome c oxidase (*COX*) DNA. Values indicate an average of 10 replications. Error bars indicate standard error.

**Figure 5 jof-07-00706-f005:**
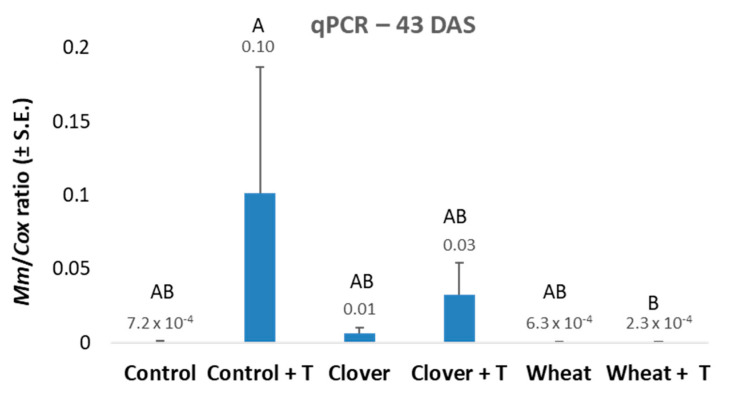
The 2020 commercial fields soil experiment—real-time PCR analysis 43 DAS. The experiment is described in Figure 2. The *y*-axis indicates *M. maydis* relative DNA abundance in the roots normalized to cytochrome c oxidase (*COX*) DNA. Values indicate an average of 10–12 replications. Error bars indicate standard error.

**Figure 6 jof-07-00706-f006:**
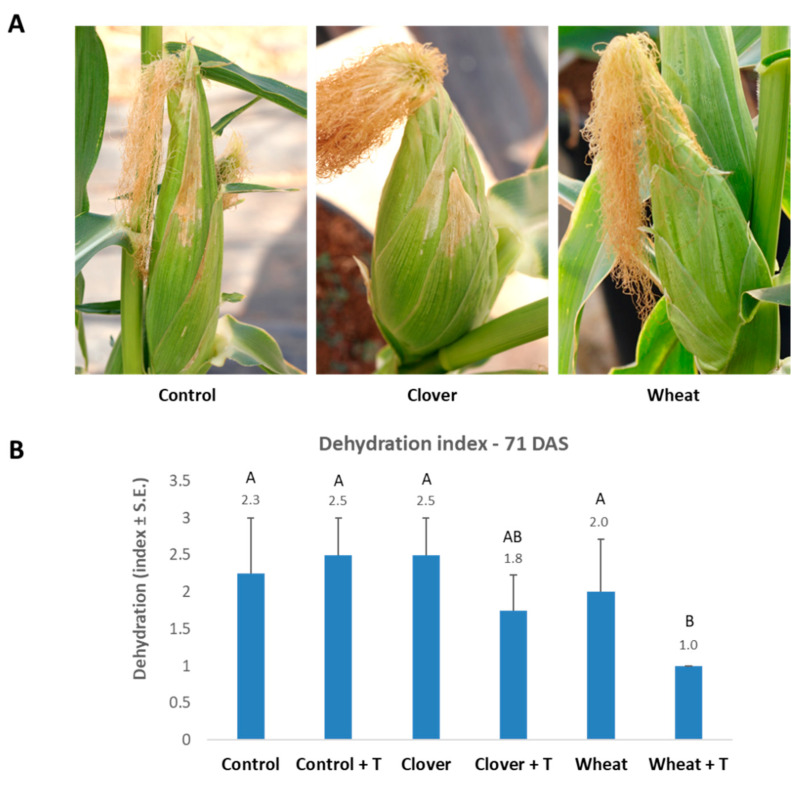
The 2020 commercial fields soil experiment—dehydration assessment on harvest day (71 DAS). The experiment is described in Figure 2. (**A**) Classification of the cobs’ spathes disease symptoms at the end of the 2020 experiment. Representative images of the treatments with varying degrees of late wilt symptoms. (**B**) Dehydration symptoms on the large bracts surrounding the cobs (the spathes) were estimated according to the following scale: 1—healthy with no signs of dehydration, 2—minor symptoms up to 20% of the cob surface, 3—moderate symptoms that cover 30–40% of the cob surface, and 4—diseased with 50% or more dehydrated cobs’ area. Values represent an average of 10–12 replications. Error bars indicate standard error. Statistically significant (one-way ANOVA, *p* < 0.05) differences between treatments at the same measures are indicated by different letters (A–B).

**Figure 7 jof-07-00706-f007:**
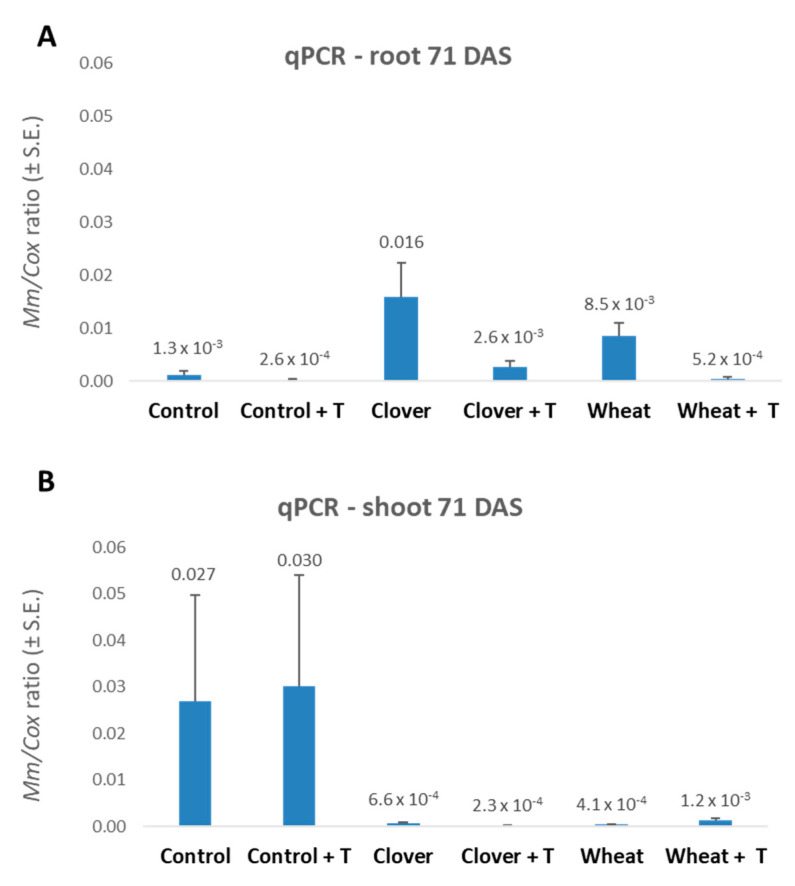
The 2020 commercial fields soil experiment—real-time PCR analysis 71 DAS. The experiment is described in Figure 2. The *y*-axis indicates *M. maydis* relative DNA abundance in the roots normalized to cytochrome c oxidase (*COX*) DNA. Values indicate an average of 10–12 replications. Error bars indicate standard error.

**Table 1 jof-07-00706-t001:** The 2019 soil microbiome integrity experiment—growth parameters at 22 DAS ^1^.

	Tillage	Emergence (%)		Root Weight (g)		Shoot Weight (g)		Leaves (No.)		Height (cm)	
Control	−	78% ± 7%	B	0.80 ± 0.07	A	10.8 ± 0.8	AB	6.7 ± 0.1	A	7.1 ± 0.3	B
Resid MG	−	94% ± 4%	A	0.88 ± 0.12	A	10.2 ± 1.5	B	5.8 ± 1.0	B	10.4 ± 0.7	A
Clover	−	84% ± 7%	AB	0.79 ± 0.08	A	13.2 ± 0.9	A	6.9 ± 0.2	A	7.4 ± 0.3	B
	+	94% ± 3%	A	0.72 ± 0.06	A	9.5 ± 1.0	B	7.3 ± 0.3	A	7.6 ± 0.3	B
Wheat	−	88% ± 5%	AB	0.85 ± 0.06	A	8.9 ± 0.6	B	5.8 ± 0.2	B	11.2 ± 0.5	A
	+	96% ± 4%	A	0.66 ± 0.08	A	10.4 ± 0.8	AB	7.0 ± 0.4	A	7.0 ± 0.4	B

^1^ Emergence percentages were measured eight days after sowing (DAS). All other data were collected after 22 days of growing maize sprouts (Prelude cv.) in a greenhouse. Values represent an average of 10 replications ± standard error. Statistically significant (one-way ANOVA, *p* < 0.05) differences between treatments at the same measures are indicated by different letters (A–B).

**Table 2 jof-07-00706-t002:** The 2019 soil microbiome integrity experiment—growth parameters at 56 DAS ^1^.

	Tillage	Phenological Development	Height (cm)	
Control	−	Dis.–1, Vt–1, R1–8	140.0 ± 9.0	AB
Resid MG	−	Dis.–2, Vt–5, R1–3	125.5 ± 9.9	BC
Clover	−	Vt–4, R1–6	117.5 ± 6.2	C
	+	Vt–1, R1–9	125.5 ± 3.2	BC
Wheat	−	R1–10	127.0 ± 3.4	BC
	+	Dead–1, Dis.–1, R1–8	147.2 ± 3.1	A

^1^ Values represent an average of 10 replications ± standard error. Dis. is a shortcut for Diseased. The maize phenological stages are described in [52]. Statistically significant (one-way ANOVA, *p* < 0.05) differences between treatments at the same measures are indicated by different letters (A–C).

**Table 3 jof-07-00706-t003:** The 2019 soil microbiome integrity experiment—growth parameters at 77 DAS ^1^.

	Tillage	Phenological Development	Shoot Wet Weight (g)		Height (cm)		Cob Wet Weight (g)	
Control	−	Vt–1, R5–9	98.1 ± 9.8	C	132.5 ± 8.6	AB	75.7 ± 6.3	B
Resid MG	−	R5–10	129.5 ± 9.4	ABC	148.8 ± 4.6	A	95.2 ± 10.6	AB
Clover	−	R5–10	116.2 ± 15.5	BC	122.5 ± 9.8	B	87.4 ± 8.5	AB
	+	V12–1, R5–9	111.7 ± 13.4	BC	122.5 ± 10.7	B	92.2 ± 11.8	AB
Wheat	−	Vt–1, R5–9	144.1 ± 10.5	AB	132.9 ± 6.0	AB	110.9 ± 9.0	A
	+	R5–10	151.4 ± 7.0	A	134.6 ± 7.0	AB	103.1 ± 11.5	AB

^1^ Values represent an average of 10 replications ± standard error. Statistically significant (one-way ANOVA, *p* < 0.05) differences between treatments at the same measures are indicated by different letters (A–C).

**Table 4 jof-07-00706-t004:** The 2019 soil microbiome integrity experiment—dry weight parameters at 77 DAS ^1^.

	Tillage	Shoot Dry Weight (g)	Cob Dry Weight (g)
Control	−	59.1 ± 6.3	47.0 ± 3.4
Resid MG	−	59.6 ± 6.1	60.8 ± 7.7
Clover	−	61.3 ± 8.3	53.9 ± 6.2
	+	47.5 ± 9.4	48.3 ± 7.6
Wheat	−	65.5 ± 3.8	61.7 ± 7.9
	+	60.4 ± 5.8	65.5 ± 6.5

^1^ Values represent an average of 10 replications ± standard error. No statistically significant (one-way ANOVA, *p* < 0.05) differences were found between the treatments or between the treatments and the control.

**Table 5 jof-07-00706-t005:** The 2020 commercial fields soil experiment—growth parameters at 43 DAS ^1^.

	Tillage	Emergence (%)	Shoot Wet Weight (g)	Leaves (no.)	Height (cm)
Control	−	58% ± 5.5%	317.2 ± 22.4	8.2 ± 0.4	78.3 ± 1.2
	+	36% ± 7.8%	307.8 ± 24.4	7.8 ± 0.4	84.0 ± 3.0
Clover	−	50% ± 10.0%	329.2 ± 20.4	8.1 ± 0.3	79.7 ± 2.3
	+	44% ± 8.3%	303.0 ± 25.5	8.1 ± 0.4	81.9 ± 2.5
Wheat	−	52% ± 9.5%	333.4 ± 17.5	8.1 ± 0.3	83.9 ± 3.5
	+	50% ± 5.4%	345.3 ± 17.6	7.9 ± 0.3	84.9 ± 1.8

^1^ Emergence percentages were measured three days after sowing. All other data were collected after 43 days of growing maize sprouts (Prelude cv.) in a greenhouse. Values represent an average of 10–12 replications ± standard error. No statistically significant (one-way ANOVA, *p* < 0.05) differences were found between the treatments or between the treatments and the control.

**Table 6 jof-07-00706-t006:** The 2020 commercial fields soil experiment—growth parameters at the season end, 71 DAS ^1^.

	Tillage	Shoot Weight (g)	Leaves (No.)	Height (cm)	Cob Wet Weight (g)	
Control	−	137.1 ± 9.4	8.7 ± 0.4	95.1 ± 4.6	118.1 ± 11.0	AB
	+	153.6 ± 9.4	8.8 ± 0.3	99.2 ± 4.2	120.3 ± 4.5	AB
Clover	−	153.4 ± 16.6	8.9 ± 0.6	100.4 ± 7.7	125.2 ± 7.7	A
	+	146.8 ± 6.6	8.1 ± 0.3	102.4 ± 2.5	122.8 ± 7.3	A
Wheat	−	147.3 ± 8.2	8.1 ± 0.2	96.9 ± 4.1	125.3 ± 9.0	A
	+	158.8 ± 10.3	8.0 ± 0.3	96.4 ± 3.1	95.5 ± 12.1	B

^1^ Values represent an average of 10–12 replications. If existing, statistically significant (one-way ANOVA, *p* < 0.05) differences between treatments at the same measures are indicated by different letters (A–B).

**Table 7 jof-07-00706-t007:** Comparative analysis of the 2019 and the 2020 experiments plants’ growth and yield results at the harvest day ^1^.

		Shoot Wet Weight (g)		Cob’s Wet Weight (g)	
	Tillage	2019	2020	2019	2020
Clover	−	118%	112%	115%	106%
	+	114%	107%	122%	104%
Wheat	−	147%	107%	146%	106%
	+	154%	116%	136%	81%

^1^ Data represent the percent difference compared to the no-tillage and no crop cycle control. Detailed data are described in Table 3 (the 2019 experiment) and Table 6 (the 2020 experiment). Harvest day was 77 DAS in 2019 and 71 DAS in 2020.

**Table 8 jof-07-00706-t008:** Comparative analysis of the 2019 and the 2020 experiments plants’ health results at the harvest day ^1^.

		Diseased Plants		*M. maydis* DNA	
	Tillage	2019	2020	2019	2020
Clover	−	65%	109%	89%	2%
	+	81%	78%	394%	1%
Wheat	−	27%	87%	18%	2%
	+	27%	43%	36%	4%

^1^ Data represent the percent difference compared to the no-tillage and no crop cycle control. Detailed data are described in Figure 3 and Figure 4 (the 2019 experiment), Figure 6B and Figure 7B (the 2020 experiment). Harvest day was 77 DAS in 2019 and 71 DAS in 2020. The *M. maydis* DNA levels in the plants’ first aboveground internode are presented. In the clover and wheat treatments, the DNA values in the 2020 experiment were similar to those of the 2019 experiment. The low DNA percentages at the 2020 experiment are because of a sharp (23-fold) elevation in the pathogen DNA in the no-tillage and no crop cycle control plants in 2020 compared to 2019.

## Data Availability

The datasets generated during and/or analyzed during the current study are available from the corresponding author on reasonable request.

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
