# Peer review of "Crop Cycle and Tillage Role in the Outbreak of Late Wilt Disease of Maize Caused by Magnaporthiopsis maydis"

_jof, 2021, doi:10.3390/jof7090706_

Round 1

Reviewer 1 Report

The present manuscript is presenting interesting findings regarding Maize Late Wilt Disease. Some corrections can be done as suggested.

  1. Provide a graphical abstract, if possible.
  2. More pictures can be added for seasonal variations.
  3. Did authors study other microorganisms present in rhizosphere?
  4. The data on seasonal studies can be better presented in a comparative way with some points in conclusion related to this.
  5. Roots appeared comparatively less affected. How it could be explained. 

Author Response

Responses to Reviewer 1’s comments

We thank the reviewer for investing substantial efforts, which are undoubtedly contributing to this manuscript. The remarks and suggestions improved this paper’s scientific soundness and accurateness. Your contribution is greatly appreciated.

Specific comments

The present manuscript is presenting interesting findings regarding Maize Late Wilt Disease. Some corrections can be done as suggested.

  1. Provide a graphical abstract, if possible.
  2. More pictures can be added for seasonal variations.
  3. Did the authors study other microorganisms present in the rhizosphere?
  4. The data on seasonal studies can be better presented in a comparative way with some points in conclusion related to this.
  5. Roots appeared comparatively less affected. How it could be explained. 

Reply to the reviewer comments

1. We agree and prepare a graphical abstract that described the experimental design and the results.

2. The dates selected for photographing the maize plants represent the main experiment stages - at sowing, at the end of the sprouting phase (30 DAS, V5 - fifth leaf stage, before the female flowering at 56 DAS), and the harvest  (71 DAS). The maize development was normal and correlated to the data in the literature (see Abendroth, L.J.; Elmore, R.W.; Boyer, M.J.; Marlay, S.K. Corn growth and development. 2011). We didn’t photograph other dates.

The following explanation was added to the text (lines 268-272): “

Seasonal variations of the main experiment stages are demonstrated in Figure 2. These stages include the sowing, the end of the sprouting phase (30 DAS, V5 - fifth leaf stage, before the female flowering at 56 DAS), and the harvest  (71 DAS). The maize development was normal and correlated to the data in the literature (see [52])”.

3. This is an excellent question. We didn’t include the study of the microorganisms present in the rhizosphere in the current work. This pioneering work aimed to uncover the potential of crop rotation and tillage on the outbreaks of late wilt disease of maize. It provides essential data that we believe will now encourage subsequent studies on the mycorrhizal and soil microflora changes regarding these agrotechnical approaches.

4. We agree, and the following changes were incorporated into the text:

  • Tabel 7 was added to the Discussion section (line 504). The table is a comparative analysis of the 2019 and the 2020 experiments plants’ growth and yield results at the harvest day.
  • Tabel 8 was added to the Discussion section (line 509). The table is a comparative analysis of the 2019 and the 2020 experiments plants’ health results at the harvest day.
  • The following paragraph was added accordingly (lines 489-502):

“A comparative analysis of the 2019 and the 2020 experiments results at the harvest day is illustrated in Table 7. When corn was sown on wheat soil, a significant improvement in the fresh weight of the shoot (47%-54% in the 2019 experiment) and cob (36%-46% in the 2019 experiment) and was achieved compared to the other treatments (clover soil, commercial mycorrhiza preparation, and the control). This achievement was not affected drastically by tillage. It was followed by a sharp decrease in disease symptoms (73% in the 2019 experiment) and the pathogen’s presence (82%-64% in the 2019 experiment) in the plants’ tissues (Table 8). Interestingly the 2020 experiment that was based on commercial fields soil resulted in more severe LWD. The disease severity is reflected in higher diseased plants percentages and sharp (23-fold) elevation in the pathogen DNA in the first aboveground internode of the no-tillage control plants (Figure 7, Table 8). This is most likely the reason for the lesser enhancement in the shoot and cobs fresh weight in the clover and wheat treatments (compared to the control) this year (Table 7)”.

5. Indeed, this is an important aspect that should be explained. The following explanation was added to the text (Lines 518-527):

“Another intriguing aspect is that at the season-ending, the roots appeared comparatively less affected by the pathogen than the shoot (Figure 7). Indeed M. maydis high DNA levels in the shoot at this growth stage (the harvest) are well documented in our previous works [3,4]. When an LWD susceptible maize cultivar was seeded on M. maydis heavily infested commercial field soli, the following changes in the pathogen’s DNA levels in the host tissues were recorded. When early signs of the disease began to appear, from day 50 onwards, the fungal DNA decreased in the roots but increased significantly in the stems until it peaked at 3.2-fold its initial level 72 d after sowing [3]. These variations in the DNA are consistent with previous literature observations on the disease’s mode and the pathogen spread from the roots to the stem, leaves, and kernels [63].

Reviewer 2 Report

In the paper entitled “Crop Cycle and Tillage Role in the Outbreak of Late Wilt Disease of Maize caused by Magnaporthiopsis maydis”,  the authors established the role of crop rotation of maize with either wheat or clover and the influence of minimum tillage in restricting maize late wilt disease caused by Magnaporthiopsis maydis. The manuscript is well written and scientific relevant and should be accepted after minor revision.

  1. The authors should check for grammar errors and paragraph structure (e.g., paragraphs consisting of a single sentence; Lines 242-243).
  2. I am worried that the Introduction section is too long?
  3. My biggest concern with the manuscript is that the authors do not discuss their work in the Discussion section. They do not contextualize their findings given what is known.

Author Response

Responses to Reviewer 2’s comments.

We want to express our sincere appreciation to the reviewer for essential and helpful advice. The time and effort invested are greatly appreciated and certainly contributed to the manuscript and improved it. Thank you.

Specific comments

In the paper entitled “Crop Cycle and Tillage Role in the Outbreak of Late Wilt Disease of Maize caused by Magnaporthiopsis maydis,” the authors established the role of crop rotation of maize with either wheat or clover and the influence of minimum tillage in restricting maize late wilt disease caused by Magnaporthiopsis maydis. The manuscript is well-written and scientific relevant and should be accepted after minor revision.

  1. The authors should check for grammar errors and paragraph structure (e.g., paragraphs consisting of a single sentence; Lines 242-243).
  2. I am worried that the Introduction section is too long?
  3. My biggest concern with the manuscript is that the authors do not discuss their work in the Discussion section. They do not contextualize their findings given what is known.

Reply to the reviewer comments

1. A professional English scientific copy editor edited the entire manuscript. Still, some typo and wording mistakes may be found. We have made our best effort to correct them.

2. The introduction section presents the current knowledge status according to the following logic: 

    • The maize late wilt disease global spreading and implications.
    • The disease causal agent, Magnaporthiopsis maydis, and the disease mode.
    • The current status of controlling the disease.
    • The problems in applying chemical options to restrict the pathogen.
    • Environmentally friendly strategies to control late wilt.
    • The current study aims and research plan.

The following paragraph was omitted from the Introduction text:

“Hence, the need to control this pathogen efficiently has become increasingly important in both commercial grain production and maize seed production. To exemplify this importance, in Israel, the extent of corn crops in metric ton yield per metric hectare exhibits a constant upward tendency, from 17.5 in 1987-1996 to 20.1 in 2007-2016 (data from the Israel Organization of Crops and Vegetables). Effective risk management of late wilt disease, mainly by avoiding the cultivation of sensitive maize hybrids, may contribute to this positive outcome.“

The following paragraph was shortened and focused:

“Maintaining the continuity and integrity of mycorrhizal networks in the soil may allow the plant to enjoy higher resistance to soil diseases [47], including late wilt disease [27]. Indeed, Arbuscular mycorrhizal fungi (AMF) have a proven ability to improve plant resistance to biotic and abiotic stresses [48]. Among other environmental factors, the local and systemic defense mechanisms in the plant are activated by the AMF in response to soil pathogens. The wide range of hosts for mycorrhizal fungi and the intact extraradical mycelium (ERM) method to maintain microbial network continuity in the soil enables the production of a crop sequence that will preserve the microbial network and help protect against soil diseases.”

The paragraph now reads:

“Maintaining the continuity and integrity of mycorrhizal networks in the soil may allow the plant to enjoy higher resistance to soil diseases [46], including late wilt disease [27]. Indeed, Arbuscular mycorrhizal fungi (AMF) have a proven ability to improve plant resistance to biotic and abiotic stresses by activated the plant’s local and systemic defense mechanisms [47]“. (lines 105-109)

We believe all other parts of the Introduction are essential to understand the current global status of late wilt disease, the knowledge gaps, and the necessity to develop new green solutions to cope with this emerging pathogen.

3. We agree. This aspect has also been raised by reviewer 2. Thus we added the following information to the text:

  • Tabel 7 was added to the discussion section (line 504). The table is a comparative analysis of the 2019 and the 2020 experiments plants’ growth and yield results at the harvest day.
  • Tabel 8 was added to the discussion section (line 509). The table is a comparative analysis of the 2019 and the 2020 experiments plants’ health results at the harvest day.
  • The following paragraph was added accordingly (lines 489-502):

“A comparative analysis of the 2019 and the 2020 experiments results at the harvest day is illustrated in Table 7. When corn was sown on wheat soil, a significant improvement in the fresh weight of the shoot (47%-54% in the 2019 experiment) and cob (36%-46% in the 2019 experiment) and was achieved compared to the other treatments (clover soil, commercial mycorrhiza preparation, and the control). This achievement was not affected drastically by tillage. It was followed by a sharp decrease in disease symptoms (73% in the 2019 experiment) and the pathogen’s presence (82%-64% in the 2019 experiment) in the plants’ tissues (Table 8). Interestingly the 2020 experiment that was based on commercial fields soil resulted in more severe LWD. The disease severity is reflected in higher diseased plants percentages and sharp (23 fold) elevation in the pathogen DNA in the first aboveground internode of the no-tillage control plants (Figure 7, Table 8). This is most likely the reason for the lesser enhancement in the shoot and cobs fresh weight in the clover and wheat treatments (compared to the control) this year (Table 7)”.

  • The following explanation was added to the text (Lines 518-527): “Another intriguing aspect is that at the season-ending, the roots appeared comparatively less affected by the pathogen than the shoot (Figure 7). Indeed M. maydis high DNA levels in the shoot at this growth stage (the harvest) are well documented in our previous works [3,4]. When an LWD susceptible maize cultivar was seeded on M. maydis heavily infested commercial field soli, the following changes in the pathogen’s DNA levels in the host tissues were recorded. When early signs of the disease began to appear, from day 50 onwards, the fungal DNA decreased in the roots but increased significantly in the stems until it peaked at 3.2-fold its initial level 72 d after sowing [3]. These variations in the DNA are consistent with previous literature observations on the disease’s mode and the pathogen spread from the roots to the stem, leaves, and kernels [63].

Reviewer 3 Report

Crop Cycle and Tillage Role in the Outbreak of Late Wilt Disease of Maize caused by Magnaporthiopsis maydis

Summary:

The maize late wilt disease (LWD) is a disease severely affecting maize fields throughout Israel. It is characterized by rather rapid dehydration of the plants near maturity and considered the most harmful disease in commercial maize fields in Israel. This paper is seeking an alternative way to cope with this fungal disease. The authors designed greenhouse experiments to test the role of crop rotation of maize with either wheat or clover and the influence of minimum tillage in restricting LWD. They also pointed out the advantages of carrying out greenhouse experiments over the field experiments. The design of the experiments is reasonable, the results are presented clearly.

Comments:

  1. The manuscript was well prepared. Nice work!
  2. Methods were described clearly, I can imagine the whole process of experiments after reading materials and methods part.

Author Response

We thank the reviewer for investing time and effort in reviewing this manuscript. Your contribution is greatly appreciated.
